# Four Chemotherapeutic Compounds That Limit Blood-Brain-Barrier Invasion by *Toxoplasma gondii*

**DOI:** 10.3390/molecules27175572

**Published:** 2022-08-30

**Authors:** Zijing Yan, Hao Yuan, Junjie Wang, Zipeng Yang, Pian Zhang, Yasser S. Mahmmod, Xiaohu Wang, Tanghui Liu, Yining Song, Zhaowen Ren, Xiu-Xiang Zhang, Zi-Guo Yuan

**Affiliations:** 1College of Veterinary Medicine, South China Agricultural University, Guangzhou 510642, China; 2Guangdong Provincial Key Laboratory of Zoonosis Prevention and Control, South China Agricultural University, Guangzhou 510642, China; 3Key Laboratory of Zoonosis of Ministry of Agriculture and Rural Affairs, South China Agricultural University, Guangzhou 510642, China; 4College of Veterinary Medicine, Xinjiang Agricultual University, Urumqi 830052, China; 5Infectious Diseases, Department of Animal Medicine, Faculty of Veterinary Medicine, Zagazig University, Zagazig 44511, Egypt; 6Veterinary Sciences Division, Al Ain Men’s College, Higher Colleges of Technology, Al Ain 17155, United Arab Emirates; 7Institute of Animal Health, Guangdong Academy of Agricultural Sciences, Guangzhou 510640, China; 8College of Agriculture, South China Agricultural University, Guangzhou 510642, China

**Keywords:** toxoplasmic encephalitis, *Toxoplasma gondii*, blood-brain barrier, SB290157, CVF, NSC23766, Anxa1

## Abstract

Background: *Toxoplasma gondii*, an intracellular protozoan parasite, exists in the host brain as cysts, which can result in Toxoplasmic Encephalitis (TE) and neurological diseases. However, few studies have been conducted on TE, particularly on how to prevent it. Previous proteomics studies have showed that the expression of C3 in rat brains was up-regulated after *T. gondii* infection. Methods: In this study, we used *T. gondii* to infect mice and bEnd 3 cells to confirm the relation between *T. gondii* and the expression of C3. BEnd3 cells membrane proteins which directly interacted with C3a were screened by pull down. Finally, animal behavior experiments were conducted to compare the differences in the inhibitory ability of TE by four chemotherapeutic compounds (SB290157, CVF, NSC23766, and Anxa1). Results: All chemotherapeutic compounds in this study can inhibit TE and cognitive behavior in the host. However, Anxa 1 is the most suitable material to inhibit mice TE. Conclusion: *T. gondii* infection promotes TE by promoting host C3 production. Anxa1 was selected as the most appropriate material to prevent TE among four chemotherapeutic compounds closely related to C3.

## 1. Introduction

*T.**gondii*, a widely spread parasitic protozoan, affects approximately one-third of the world’s population [1]. *T. gondii* can use macrophages and dendritic cells to reach various organs through host blood circulation [2]. *T. gondii* can break through the host blood–brain barrier (BBB) to infect the brain tissue [3] and activate microglia and astrocytes. Mouse brain showed signs of inflammation ten days after infection, which is the process of the generation of Toxoplasma encephalitis (TE) [4]. The host, with the help of the pro-inflammatory cytokine interferon-gamma (IFN-γ), peripheral monocytes, and T cells, controls and eliminates parasites in the brain [5,6]. Nevertheless, due to the inhibitory effect of the host immune system on *T. gondii*, healthy hosts show mild or asymptomatic infection. Thus, *T. gondii* usually exists in the form of cysts in the brain tissue of a chronically infected host [1].

When *T. gondii* patients get infected with HIV, cancer, and other immunosuppressive diseases, stable *T. gondii* cysts in the brain will be activated and reactivate TE [7]. Previous studies found that animal models which are chronically infected with *T. gondii* showed behavioral and cognitive abnormalities [8]. For instance, the attitude of rodents infected with *T. gondii* towards predator’s odor has changed from rejection to favor [9,10]. Moreover, other studies found that the seropositive rate of *T. gondii* was positively correlated with the occurrence of bipolar disorder [11] and dementia [12]. These examples demonstrate that there is a strong relation between *T. gondii* infection and mental illness. Most drugs for *T. gondii* such as sulfadiazine, spiramycin, pyrimidine, and clindamycin are used to eliminate tachyzoites [13]. However, it is urgent to identify new therapeutic methods and drug targets for *T. gondii* chronic infection [14].

The complement system has many important roles, including phagocytizing foreign bodies, clearing immune complexes, and apoptotic cells, and participating in acquired immunity as a part of the innate immune system. Three activation pathways of complement can produce C3 convertase and promote resolution of C3 into the effector molecules C3a and C3b [15]. C3a is an allergic toxin which can promote an inflammatory response and the proliferation of glial cells. However, the function of C3a is far more extensive than these, and the exploration of C3a roles still needs to be studied in-depth [16]. C3 is closely related to the proliferation of *T. gondii* in the host and the occurrence of complications. It has been proved that *T. gondii* can recruit C4BP and factor H from the host, inactivate the surface-bound C3 and limit C5b-9 from forming a membrane attack complex [17,18], eventually resulting in immune escape. It was reported that the complement pathway and tight junction (TJ) pathway have specific changes in mouse brain tissues because of the significant increase in C3 expression led by *T. gondii* infection [19]. C3 may be related to the process of *T. gondii* entering the host brain. Therefore, we can start from the direction of regulating C3 expression to prevent TE.

Complement inhibitors can be divided into three categories: firstly, chemotherapeutic compounds can be used to inhibit the activity of components in the complement activation pathway, such as compstatin, a cyclic 13 residue peptide, which binds with C3 to inactivate it [20]. Secondly, chemotherapeutic compounds can prevent the components of an activating complement from acting, such as C3aR inhibitors, which aim to inhibit the pro-inflammatory activity of C3a [21]. Lastly, chemotherapeutic compounds can be used to consume complement components, such as cobra venom factor (CVF). CVF, a non-toxic protein purified from cobra venom, can produce C3/C5 invertase to deplete C3/C5 in the host [22,23]. CVF has been proved to consume the complement safely and it is often used to make animal models in the experiment of organ transplantation lacking complement [24,25]. SB290157, N(2)-[(2,2-diphenylethoxy) acetyl]-L-arginine, was firstly reported as a competitive C3aR inhibitor in 2001 [26], which is a conventional drug frequently used in exploring C3a/C3aR in recent years. SB290157 has been proved to improve the survival rate of patients with experimental lupus nephritis [27] and reduce the infarct volume in the thrombosis stroke model [28]. NSC23766, an inhibitor of Rac1 protein and NO synthase, often used in cytoskeleton research [29], has been proved to inhibit the invasion and replication of *T. gondii* in the parasitophorous vacuole (PV) [30].

BBB, a transcellular barrier composed of BMECs, makes up of adjacent cells which have TJ [31,32]. BBB restricts substances from crossing the central nervous system (CNS) and maintains the CNS in a stable state. After comparing the expression levels of TJ proteins (Claudin-5, Occludin and ZO-1) from brain-derived endothelial cells 3 (bEnd3), brain-derived endothelial cells 3 (bEnd5) and mouse brain endothelial cells 4 (MBEC4), previous research found that bEnd3 cells were the most suitable model for evaluating BBB function [33]. BEnd3 cells were used to study the effect of ME49 infection on C3a expression. Claudin-5 is essentially a mechanical connection between single endothelial cells, and there is a high resistance between 1500~2000 Ω·cm^2^ [34,35]. The BBB of Claudin-5 knockout mice allowed tracers with a molecular weight of less than 800 Da to exude from the blood vessels, but the mice would die within 10 h after birth [36]. ZO proteins, a transmembrane junction protein, are composed of ZO-1, ZO-2, and ZO-3 [37]. ZO proteins can not only regulate the role of the cell barrier bypass pathway, but also interact with cytoplasmic proteins to recruit various proteins involved in signal transduction and transcriptional regulation [38].

In this study, we used Western blotting, ELISA, and quantitative PCR to determine whether *T. gondii* infection can promote the expression level of C3a in host cells and then affect the expression of TJ and the permeability of the BBB. Next, to clarify the relation between C3a and BBB proteins, we used pulldown and screened Annexin A1 (Anxa1), which is closely related to C3a. Finally, we proved that the four chemotherapeutic compounds (SB290157, CVF, NSC23766, and Anxa1) could inhibit TE and improve the cognitive behavior of infected mice by using quantitative PCR and animal behavior experiments. Collectively, our study offers the first insights into a new choice for C3a closely related protein and provides a new concept for the prevention and treatment of TE.

## 2. Results

### 2.1. T. gondii Increased C3 Expression and BBB Disruption

ELISA was used to determine the changing trend of C3a in the blood circulation of mice after *T. gondii* infection and then to observe the infection on BBB caused by the changes of C3a. We observed that *T. gondii*-infected groups had peaks and C3a-addition led to peak advance (Figure 1A). TEM showed that the brain microvascular endothelial cells (BMECs) deformed and presented tail-like protrusions (Figure 1B) during the peak period of the ME49 and C3a group (144h). These results suggest that *T. gondii* infection leads to C3a increase in the host blood and disrupts BBB during the peak period of C3a.

To clarify the relation between *T. gondii* and C3a expression, we next decided to perform studies about different infection time and multiplicity of infection (MOI) of *T. gondii*. In connection with that, we found that C3a contents of the bEnd3 cells infected with *T. gondii* increased with the length of infection time (Figure 1C) and peaked when MOI was 2 and recovered when MOI was 10 (Figure 1D). However, the expressions of iNOS and Nitric oxide were always lower than that of the control group and the trends were the same as that of C3a (Figure 1E,F). This finding indicates that *T. gondii* infection can increase C3a expression in bEnd3 cells and that the up-regulation is affected by infection time and MOI.

The next question was about the relation between C3a concentration and BBB integrity. iNOS was inhibited with the ascending concentration of C3a (Figure 1G). Meanwhile, the expressions of ZO-1 and Claudin-5 were lower than the control group (Figure 1H,I). These results suggest that C3a inhibits TJ proteins (ZO-1 and Claudin-5) expression in bEnd3 cells and destroys BBB.

### 2.2. SB290157 Combined with C3a at Low Concentration Inhibited the Expression of C3a, iNOS, and Vimentin

SB290157 is a common C3aR antagonist which can inhibit C3a by competitive binding with C3aR [27]. Vimentin is the main component of the cytoskeleton and significantly rearranges around PV during *T. gondii* infection [39]. To further observe the relation between C3a and BBB permeability, statistical analysis was done by comparing the ME49, different concentrations of C3a and SB290157 groups with the control group. Interestingly, ME49, C3a (0.01 μg and 0.05 μg), and SB290157 can promote the expression of C3a and iNOS in bEnd3 cells compared with the control group. BEnd3 cells treated with a low concentration of C3a (0.01 μg) had higher expression of C3a, iNOS, and Vimentin than bEnd3 cells treated with a high concentration of C3a (0.05 μg). In addition, compared with the group that added 0.01 μg C3a, the group C3a combined with SB290157 had low expression of C3a, iNOS, and Vimentin (Figure 2A–C).

### 2.3. NSC23766 Promoted TJ Proteins Express

NSC23766 is closely related to the cytoskeleton, thus inhibiting the invasion and proliferation of *T. gondii.* NSC23766 can inhibit the activation of Rac1 protein and NO synthase. Activating of Rac1 will cause the reorganization of the host cytoskeleton and promote the invasion of *T. gondii*. After interfering with or silencing the Rac1 gene, the infection rate of *T. gondii* is significantly reduced [29]. Compared with the group only infected ME49, C3a with ME49 group, and CVF with ME49 group could inhibit the expression of Vimentin in bEnd3 cells. But the ME49 with NSC23766 group could inhibit the expression of C3a and Vimentin and promote the expression of TJ proteins (ZO-1 and claudin-5). The addition of C3a or CVF promoted the expression of ZO-1 in the NSC23766 group but had no significant effect on the expression of Claudin-5. Compared with the ME49 group, the expression of ZO-1 and Claudin-5 was significantly increased in the ME49 and NSC23766 groups (Figure 2D–G).

### 2.4. SB290157 and NSC23766 Inhibited T. gondii Invasion into bEnd3 Cells

Compared with the control group, the expression of Claudin-5 in the ME49-mCherry group was decreased. The addition of C3a inhibited the expression of Claudin-5 and promoted the invasion of ME49-mCherry. The addition of SB290157 did not affectthe expression of Claudin-5, but it could inhibit the invasion of ME49-mCherry. The addition of NSC23766 reduced the decrease of Claudin-5 and the invasion of ME49-mCherry (Figure 3). Our findings show that SB290157 and NSC23766 can inhibit the invasion of *T. gondii* into the bEnd3 cells with different methods.

### 2.5. Anxa1 Was Picked out as a Protein Which Could Directly Act on C3a

The lysate of bEnd3 cells was used to screen membrane proteins which can interact directly with C3a to further explore the relation between C3a and bEnd3 cells. There were many obvious protein bands revealed by silver staining in lanes 3 and 6, which could act directly with GST-C3a (Figure 4A). By comparing the information of all proteins directly interacting with GST-C3a, 141 different proteins were obtained. Except for the same proteins as those in control group 1 and 2, there were 77 unique proteins in the experimental group (Figure 4B). The unique proteins of the experimental group were involved in cellular process, metabolic process, cellular anatomical entity, intracellular, binding, and other functions, and there were 14 proteins with structural molecular activity (Figure 4D). Anxa1, P4hb, and Ddx41 all have a direct relation with C3 (Figure 4C).

### 2.6. Four Chemotherapeutic Compounds Inhibited T. gondii from Entering the Mice Brains, Protecting Mice from Mental Cognition Impairment

The transcription level of GRA7 (specific protein of *T. gondii*) was detected in brain tissue and blood of infected mice to compare the inhibitory effects of four chemotherapeutic compounds (SB290157, CVF, NSC23766, and Anxa1) on TE. Dense granule antigen (GRA) is a secretory vesicular organelle in *T. gondii*. GRA7 is a target antigen in the intracerebral immune response during the chronic phase of infection among the GRA proteins [40]. The expression of GRA7 in the four chemotherapeutic compounds groups decreased in brain tissue, which was a significant difference from the ME49 group. The GRA7’s transcription levels following treatments with SB290157 and Anxa1 in brain tissue were the lowest and there was no significant difference in-between. The comparison among the four experimental groups, shown in Figure 5A, reveals that GRA7 mRNA presented a higher accumulation after exposure to NSC23766.

The GRA7’s expression level after treatments with SB290157 and Anxa1 showed strong inhibitory effects, and the inhibition ability of groups treated with Anxa1 and NSC23766 was relatively weak. However, there were significant differences on GRA7’s expression between the ME49 group and ME49 with four chemotherapeutic medication groups in blood samples (Figure 5B). These results showed that the four chemotherapeutic compounds have an obvious effect on inhibiting the entry of *T. gondii* into the brain tissue.

For further exploration of the difference in cognitive behavior induced by inhibiting ME49 entering into the brain among mice treated with four chemotherapeutic compounds, we used a rotating rod instrument and Morris Water Maze to detect the changes in memory and balance in the mice. The residence time of mice in the ME49 group was significantly lower than the control group and there was no significant difference among the groups treated with SB290157, CVF, Anxa1, and the control group in the rotating rod instrument experiment (Figure 5C). There was a significant difference in the time spent in finding the platforms between the ME49 group and the control group during the water maze experiment, but no significant difference among the four chemotherapeutic compounds groups and the control group (Figure 5D). *T**. gondii* reduces the expression of TJ proteins in the second rectangle, SB290157 and CVF can reduce the inhibition of *T**. gondii* on TJ proteins by acting on C3a and C3aR in the third rectangle, NS23766 and Anxa1 can affect the cytoskeleton to alleviate the inhibition of *T**. gondii* on TJ proteins in the fourth rectangle (Figure 5E). Together, these results provided the important insight that the four chemotherapeutic compounds could inhibit the cognitive impairment of mice by inhibiting *T. gondii* entry into the brain.

## 3. Discussion

There was a significant positive correlation between ME49 infection time and C3a expression. ME49 invaded cells in 4 h and mainly played the role of replication in cells in 16 h. This result could be explained by the fact that C3 is expressed in the process of ME49 invasion and copy (Figure 1C). With the improvement of MOI, the C3a-expression trend increased first (MOI:0–2) and then decreased slightly (MOI:2–10). This change could be attributed to the activity of bEnd3 cells. When bEnd3 cells were infected with a large number of *T. gondii*, which impaired the activity of cells and made the cells type atrophy, the amount of C3a secreted by the cells is reduced. Infection with fewer *T. gondii* causes less damage to bEnd3 cells, and the expression of C3a increases after the more vigorous cells are stimulated. This is why the C3a expression in Figure 1D had a peak when MOI was 2. The above results showed that *T**.gondii* could promote the secretion of C3a in bEnd3 cells without affecting cell activity. This was consistent with the significant increase of C3 expression in the mice brains infected with *T. gondii* [19]. Arginine can induce tachyzoites translated into bradyzoites during the conversion process of iNOS to NO [41]. *T. gondii* infection promoted C3a expression and inhibited TJ proteins at the cellular level (Figure 1C–F). In order to further explore the effect at the host level, we found that the ME49 group could produce a peak of C3a in the systemic circulation of the mouse at 192 h, the addition or depletion of C3a advanced this peak (Figure 1A). Electron microscopy could observe the endothelial cell deformation caused by different concentrations of hypertonic solutions, which would finally lead to the opening of TJ [42]. We found structural disruption of BBB cells during the peak period of host C3a expression (Figure 1B). The above results showed that *T. gondii* infection promoted the expression of C3a in the host, inhibited the expression of TJ, led to the destruction of BBB, and then promoted its entry into the brain. Previous studies had used a simple diffusion system to deliver drugs through BBB [43]; however, we were able to transport drugs into the brain during the brief period of BBB damage mentioned above.

Three proteins: Anxa1, Ddx41, and P4hb, were obtained by pulling down of C3a and bEnd3 cells (Figure 2C). Anxa1 is an anti-inflammatory factor, which can protect neurons from ischemic injury [44,45]. ANXA1 binds to negatively charged phosphatidylserine (PS) to induce membrane crosslinking and promote membrane fusion repair when activated by micromolar Ca^2+^. This process is an important function of Anxa1 to maintain membrane integrity during membrane damage [46,47]. Ddx41, a member of the DEAD-box gene family, is an intracellular DNA sensor in medullary dendritic cells which can induce the expression of type I interferon and inhibit tumors [48]. P4hb is a protein disulfide isomerase, which can catalyze the formation and rearrangement of disulfide bonds. P4hb can increase the occurrence of reactive oxygen species [49], bladder cancer [50,51], prostate cancer [52], brain, CNS, and other related cancers. Previous pulldown studies on BBB mainly focused on the relation between brain tissue and protein in rats. Wang Z studied the interaction between brain tissue and GST-Rac1 protein in the area of intracerebral hemorrhage in SD rats [53]. Other research investigated the interaction between cell lysates of astrocytes, microglia, and SNHG5 [54]. This study is the first pulldown exploration of bEnd3 cells as the representative of BBB and the first screening of C3 and bEnd3 cell interaction proteins.

In this study, the research direction of four TE chemotherapeutic compounds were to prevent *T. gondii* from breaking through BBB into the brain tissue. We compared the protective effect between CVF (C3a depleted) and SB290157 (C3aR inhibitor). In terms of cell structure, NO could regulate the TJ of the intestinal barrier composed of intestinal epithelial cells by changing the balance of glutathione [55]. Thus, we selected NSC23766 because it is often used in NO related research and Anxa1 because it was screened by pulldown to inhibit BBB injury.

Compared with the mice from the untreated group which was infected with *T. gondii*, the mice from the *T. gondii* infection group with the addition of four chemotherapeutic compounds were able to stay on the rotating rod instrument for a longer time and spend less time in the water maze to find the platforms. These results are more similar to the control group without *T**.gondii* infection. The above results showed that the cognitive learning ability of the *T. gondii* infected group was improved after the chemotherapeutic compounds treatment (Figure 5C,D).

Compared to the group only infected with *T. gondii* and the *T. gondii* infection groups treated with NSC23766 and Anxa1, the content of *T. gondii* in the blood of the *T. gondii* infected group treated with CVF and SB290157 decreased more significantly. The addition of CVF and SB290157 reduced the effect of C3a, which might allow reduction of C3 and inhibit the proliferation of *T. gondii* in the blood. (Figure 5B). However, long-term use of SB290157 and CVF might cause chronic low levels of C3a in the host blood, which poses a threat to host health. Previous studies have showed that SB290157 plays a role as a C3a antagonist in primary cells, but has an effective C3a agonist activity in transfected cells. Furthermore, it also played a part in C5aR2 agonist activity at a higher concentration [56]. The expression of C3a in the SB290157 group was higher than that in the control group (Figure 2A). Hence, SB290157 and CVF are not good choices for controlling TE.

SB290157, CVF, and Anxa1 showed a stronger ability to prevent *T. gondii* from entering the brain than NSC23766 (Figure 5A). Since its discovery, Anxa1 has been proved to play a role in blocking leukocyte extravasation, inducing apoptosis, regulating cytokine synthesis [57], repairing the integrity of the cell membrane, and regulating the tightness of the blood-brain barrier (BBB) [58]. *T.gondii* tachyzoites converge in the BBB before invading the CNS and invade the host CNS from the intercellular space by destroying the TJ between cells and using their actin and myosin to complete the “sliding movement” [59,60]. Anxa1 plays an antiparasitic and anti-inflammatory role in the placenta infected with *T. gondii*. Compared with placentas in the first trimester expressing more Anxa1, placental explants in the third trimester express less Anxa1 and are more likely to be infected with *T. gondii* [61]. In summary, Anxa1 was the best choice for restraining TE by promoting the TJ of BBB. *T. gondii* can promote the expression of C3 and affect the TJ of BBB. Anxa1, which interacts directly with C3, can repair the TJ of the host BBB, thereby inhibiting *T. gondii* from invading the CNS from the cell bypass. However, Anxa1 only affects the TJ of cells and has no apparent inhibitory effect on the proliferation of *T**. gondii* in the host. The relation between Anxa1 and C3 has only been proved to work directly, but the mechanism of action, such as specific action time and measurement needs further discussion.

## 4. Materials and Methods

### 4.1. Experimental Animal

Kunming mice (KM), half male and half female, 4–6 weeks old, were purchased from Guangdong Experimental Animal Center in China. Mice were raised under specific pathogen-free standard conditions with stable temperature (22 ± 1 °C), 50 ± 10% humidity, free access to food and water, and a 12/12-h light–dark cycle. All operations were approved by the experimental animal ethics committee (2019f163) of South China Agricultural University and met the international requirements on animal experimental welfare.

### 4.2. Chemotherapeutic Drugs

Among four chemotherapeutic compounds, only SB290157 (1 μM, added 1 h prior to *T. gondii* infected, Sigma, Burlington, MA, USA) dissolved in DMSO, whereas the others were diluted with phosphate buffer saline (PBS) or 0.9% saline. CVF (0.01 μg/mL, Biorbyt, Cambridge, UK), NSC23766 (1 μM, added 12 h prior to *T. gondii* infected, Top science, Shanghai, China) were dissolved in PBS during *T. gondii* infected cells experiments in vitro. The animal experiments used Anxa1 (Abcam, Cambridge, UK) and other drugs diluted in 0.9% saline (7.5 μg/mL, 0.2 ml each mouse, added 12 h after *T. gondii* infected).

### 4.3. Parasites and Cells

*T. gondii* (strain ME49, genotype #2, preserved in South China Agricultural University) was cultured with Human foreskin fibroblasts (HFFs) in Dulbecco Modified Eagle Medium (DMEM) with 2% fetal bovine serum (FBS). ME49-mCherry was the same as above. Brain-derived Endothelial cells 3 (bEnd 3) cells of mouse origin (preserved in South China Agricultural University) were cultured under standard conditions using DMEM with 10% FBS at 37 °C and 5% CO^2^.

### 4.4. ELISA and Electron Microscopy

Thirty mice were randomized and treated with 0.9% saline (Control), ME49 cysts (ME49), ME49 cysts and C3a (ME49 + C3a), ME49 cysts and CVF (ME49 + CVF), and CVF (CVF). Me49 cysts were used to infect mice by gavage 5 cysts each mouse. Blood samples were collected by each group from 24 h before infection until 336 h after infection with 24 hours’ time intervals. The mouse C3a ELISA kit (Abcam, Cambridge, UK) was used for sample detection. All steps were in accordance with the developers’ instructions.

The parietal cortex tissue obtained from the control and ME49 with C3a groups during 96 to 168 h were preserved in 1% glutaraldehyde at 4 °C. After overnight treatment with 1% osmic acid, gradient dehydration was carried out with different concentrations of ethanol and 100% acetone. The sections were stained with uranyl acetate and lead citrate and then dried. The samples were placed in the transmission electron microscope (TEM) (Thermo Fisher, Talos l120 c, Waltham, MA, USA) to observe morphological changes in blood vessels.

### 4.5. Western Blotting

RIPA (Beyotime, Shanghai, China) and PMSF (Beyotime, Shanghai, China) were used to cleave bEnd3 cells to obtain proteins. The proteins were separated on a polyacrylamide gel and transferred to PVDF Western Blotting Membranes (Millipore, Burlington, MA, USA). After blocking, membranes incubated with primary antibodies overnight at 4 °C. The antibodies were included anti-C3 antibody (rabbit, 1:100, Abcam, Cambridge, UK), anti-Claudin-5 antibody (mouse, 1:100, Thermo Fisher, Waltham, MA, USA), and anti-ZO-1 antibody (rabbit, 1:100, Thermo Fisher, Waltham, MA, USA). Afterward the horseradish-peroxidase-conjugated secondary antibody was added and incubated on the shaking table for 1 h at 37 °C. The proteins were observed by an extremely highly sensitive chemiluminescence Kit (Beyotime, Shanghai, China). Using ImageJ to perform densitometry analysis.

### 4.6. Measurement of Nitric Oxide

Nitrite concentration was assessed using the Griess reaction. Griess Reagent (Beyotime, Shanghai, China) and the culture medium of cells were reacted for 10 min at room temperature. Measured absorbance was done at 550 nm and calculated using a sodium nitrite standard curve.

### 4.7. Pull down and LC-MS

The constructed prokaryotic expression vector of the GST-C3a vector was constructed by pGEX-4T-1 and C3a (HM105585) and the GST tag was added to the vector by primers. GST-C3a was transferred into *Escherichia coli* BL21 with 0.1 mM IPTG (Sigma, Burlington, MA, USA) which can induce the expression of GST-C3a protein. GST-C3a protein was obtained by protein lysate (Fitgene, Guangzhou, China) with ultrasound (1 s for each round, 1 s for each interval and 5 min for total ultrasound time) and the membrane proteins of bEnd3 cells were extracted by the membrane protein extraction kit (Thermo Fisher, Waltham, MA, USA). Pull down was divided into three groups: GST and membrane proteins of bEnd3 cells (control group 1), GST-C3a and lysis buffer (control group 2), and GST-C3a and membrane proteins of bEnd3 cells (experimental group). The resin responded with bacteria lysis solution first, then reacted with membrane proteins or lysis buffer. Finally, it was eluted with 20 mM reduced glutathione (Sigma, Burlington, MA, USA) to obtain pulldown products. The peptides separated by chromatographic column were monitored by Thermo Scientific^TM^ Q Exactive^TM^ (Thermo Scientific, Waltham, MA, USA). The original data were processed by MM File Conversion and annotated by the UniProt database. With the help of the protein function analysis database (QuickGO), metabolic pathway database (Pathway), and protein interaction database (STRING-DB), bioinformatics analyses were carried out on the target proteins to select the most suitable protein.

### 4.8. Immunofluorescence

ME49-mCherry and bEnd3 cells were cultured on coverslips, fixed with 4% paraformaldehyde for 10 min, reacted with 0.2% Triton X-100 for 5 min, and finally blocked with goat serum for 1 h. Anti-Claudin-5 (mouse, 1:100, Thermo Fisher, Waltham, MA, USA) reaction at 4 °C overnight. FITC-conjugated anti-mouse IgG (Goat, 1:500, Abcam, Cambridge, UK) were incubated for 1 h. Coverslips were stained with DAPI (Sigma, Burlington, MA, USA) for 5 min and observed with the fluorescence microscope (Leica, Weztlar, Germany).

### 4.9. Quantitative PCR

Thirty-six 8-week-old mice were randomly divided into six groups. The control group of animals was administered with 0.9% saline and the other groups of animals were treated with ME49 cysts (ME49), ME49 cysts and CVF (ME49 + CVF), ME49 cysts and SB290157 (ME49 + SB290157), ME49 cysts and NSC23766 (ME49 + NSC23766), and ME49 cysts and Anxa1 (ME49 + Anxa1).

Total RNA of the cells was extracted with TRIzol reagent (TaKaRa, Kusatsu, Japan) from the brain tissues and blood samples of mice. After obtaining the sample cDNA by RT Super Mix for quantitative PCR (Vazyme, Nanjing, China), the cDNA, primers, water, and SYBR quantitative PCR Master Mix (Vazyme, Nanjing, China) were mixed in proportion, and the programs were run by using the lift cycle 96 of fluorescence quantitative PCR instrument, with 40 cycles. All steps were in accordance with the developer’s instructions. Actb was used as a housekeeping gene to calculate the expression fold-change. The primer sequences (Table 1) were synthesized by Sangon Biotech Company (Shanghai, China).

### 4.10. Rotarod Test and Morris Water Maze

The mice were trained to adapt to the Rotarod test after 15 days of infection. Rotarod was turned on and mice were made to walk on the rotarod without falling when the adaptation training was completed. The rotating speed of the instrument was 40 rpm per min, each experiment was 10 min, and each mouse had three chances for adaptation and test. The latency to fall of the mice in each group was recorded and compared.

The round water pool was divided into four quadrants for the Morris water maze test. Mice were trained to swim to find escape platforms for 4 days before the experiment. At the beginning of the formal experiment, the escape platform was fixed in the NW quadrant, water temperature was kept at approximately 25 °C, changes to the surrounding environment and the reference object were minimized to avoid experimental error. Each mouse had four chances to enter water from different quadrants during the experiment. The time for each group to find the platform within 60 s was recorded and compared. Platforms were searched successfully if mice stayed on the platforms for more than 3 s during the period. The time from the mouse entering the water to the successful platform search was recorded as the latency.

## 5. Conclusions

In conclusion, our study found that *T. gondii* infection could increase C3, which might be related to BBB damage. Anxa1 protein, which interacted directly with C3a as bEnd3 cells protein, played a better role in preventing TE than SB290157, CVF, and NSC23766. Anxa1 could be further studied as a preventative agent for TE.

## Figures and Tables

**Figure 1 molecules-27-05572-f001:**
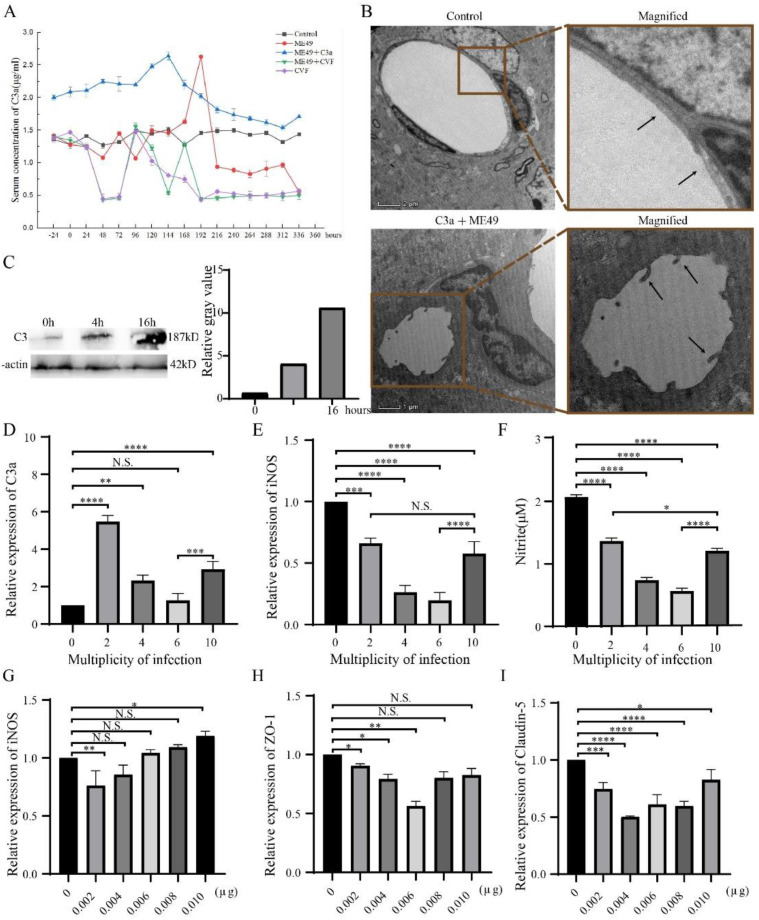
*T. gondii* promoted the expression of C3a and induced the disruption of BBB as observed by TEM analysis of BMEC. (**A**) Change trends of C3a in the blood of mice treated with ME49, C3a and CVF from –24 to 336 h. (**B**) TEM photos of the control group and C3a + ME49 at 144h. Magnification: control, denomination: 6700×; C3a + ME49, denomination:3400x. (**C**) The expression of C3 in bEnd3 cells with increasing *T. gondii* infection time. (**D**–**F**) Expression changes of C3a, iNOS and Nitrite in bEnd3 cells with different MOI of T. gondii. Using β-Actin to normalized (**D**,**E**). (**G**–**I**) Expression changes of iNOS, ZO-1, and Claudin-5 in bEnd3 cells with different concentrations of C3a. β-Actin is used to normalize. Statistical differences were analyzed by one-way ANOVA. N.S. *p* ≥ 0.05, * *p* < 0.05, ** *p* < 0.01, *** *p* < 0.001, **** *p* < 0.0001.

**Figure 2 molecules-27-05572-f002:**
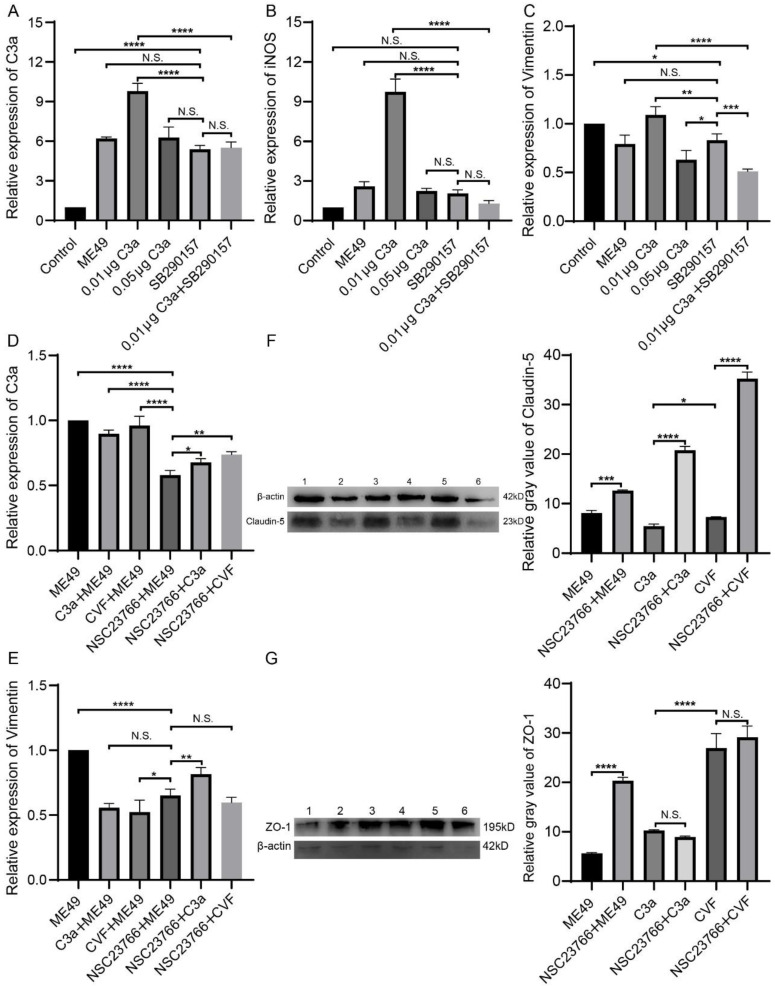
The addition of SB290157 inhibited the promoting effect of low concentration C3a on the expression of C3a, iNOS, and Vimentin in bEnd3 cells, the addition of NSC23766 inhibited the promoting effect of ME49 on the expression of C3a and Vimentin and promoted the expression of TJ proteins (ZO-1 and Claudin-5). (**A**–**C**) The expression of C3a, iNOS, and Vimentin in bEnd3 cells which were treated with ME49, C3a, and SB290157, the relative expressions were used β-Actin to normalized. (**D**–**G**) The expression effect of C3a, Vimentin, and TJ proteins in bEnd3 cells which were treated with ME49, C3a, CVF, and NSC23766. The gene expressions were determined by quantitative PCR (**D**,**E**), and the relative expressions were normalized by β-Actin. The protein expressions were measured by Western blotting. Gray values were analyzed by ImageJ software. (**F**,**G**). Statistical differences were analyzed by one-way ANOVA. N.S. *p* ≥ 0.05, * *p* < 0.05, ** *p* < 0.01, *** *p* < 0.001, **** *p* < 0.0001.

**Figure 3 molecules-27-05572-f003:**
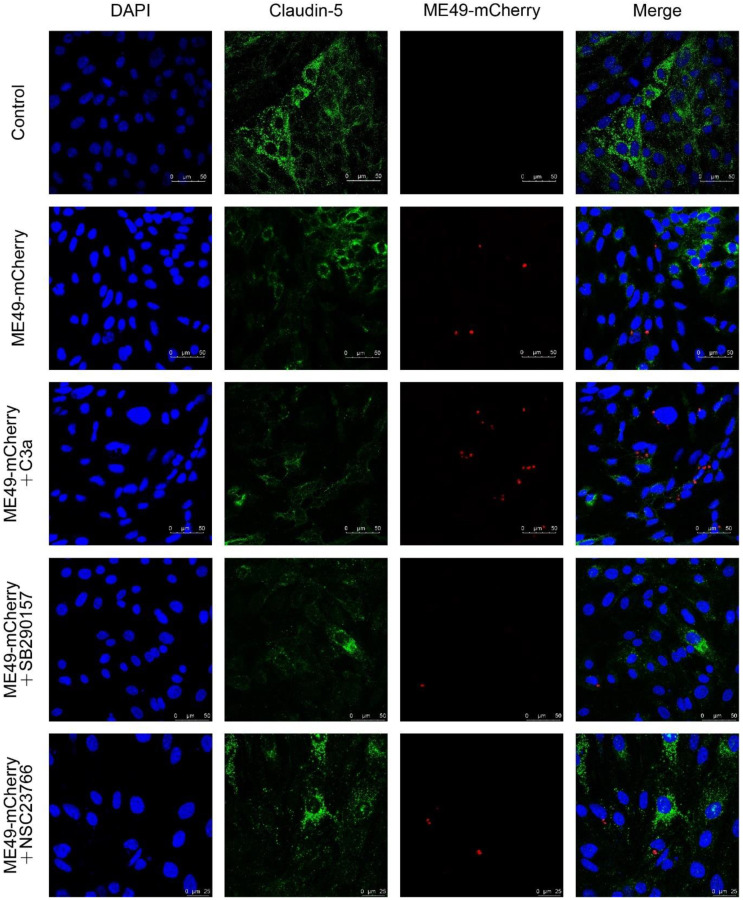
SB290157 and NSC23766 restrained Claudin-5 expression and inhibited *T. gondii* entering into bEnd3 cells. Representative images of Claudin-5 and ME49-mCherry immunofluorescence staining (200×). Blue, DAPI staining of DNA; Green, Claudin-5; Red, ME49-mCherry.

**Figure 4 molecules-27-05572-f004:**
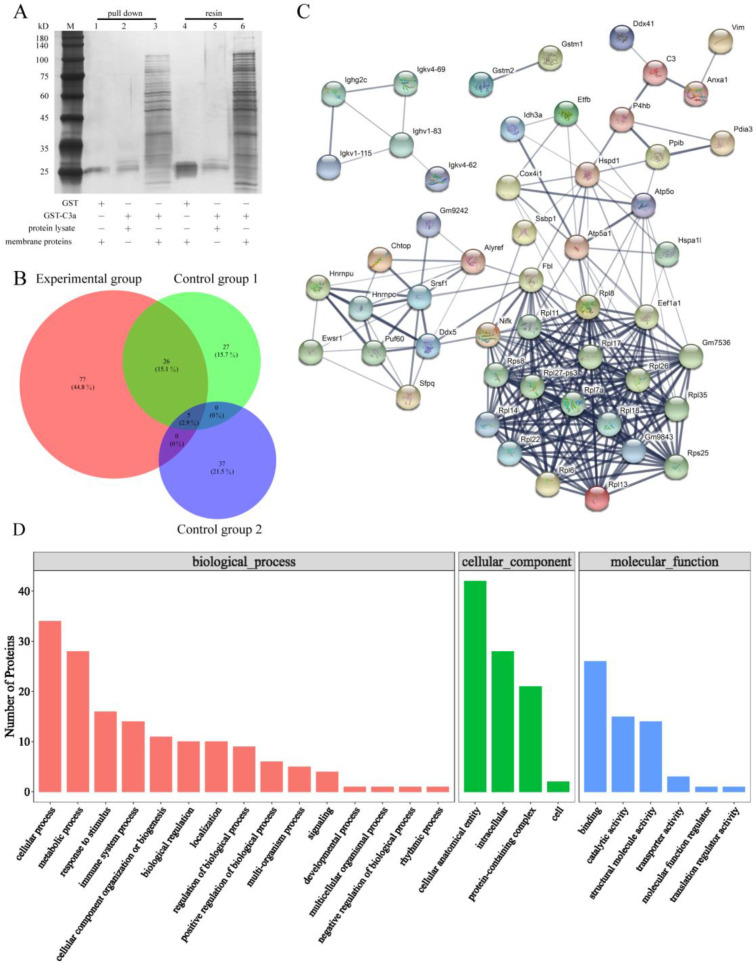
Anxa1 was screened as a protein that directly act on C3a. (**A**) Silver staining via Flag-tagged pulldown confirmed that C3a interacted with various membrane proteins from bEnd3 cells. (**B**) The Wayne figure of proteins in control group 1, control group 2 and experimental group. (**C**) STRING interaction network of the unique proteins of the membrane proteins from bEnd3 cells which was derived from the STRING database. (**D**) GO functional enrichment analysis of the unique proteins of the membrane proteins from bEnd3 cells.

**Figure 5 molecules-27-05572-f005:**
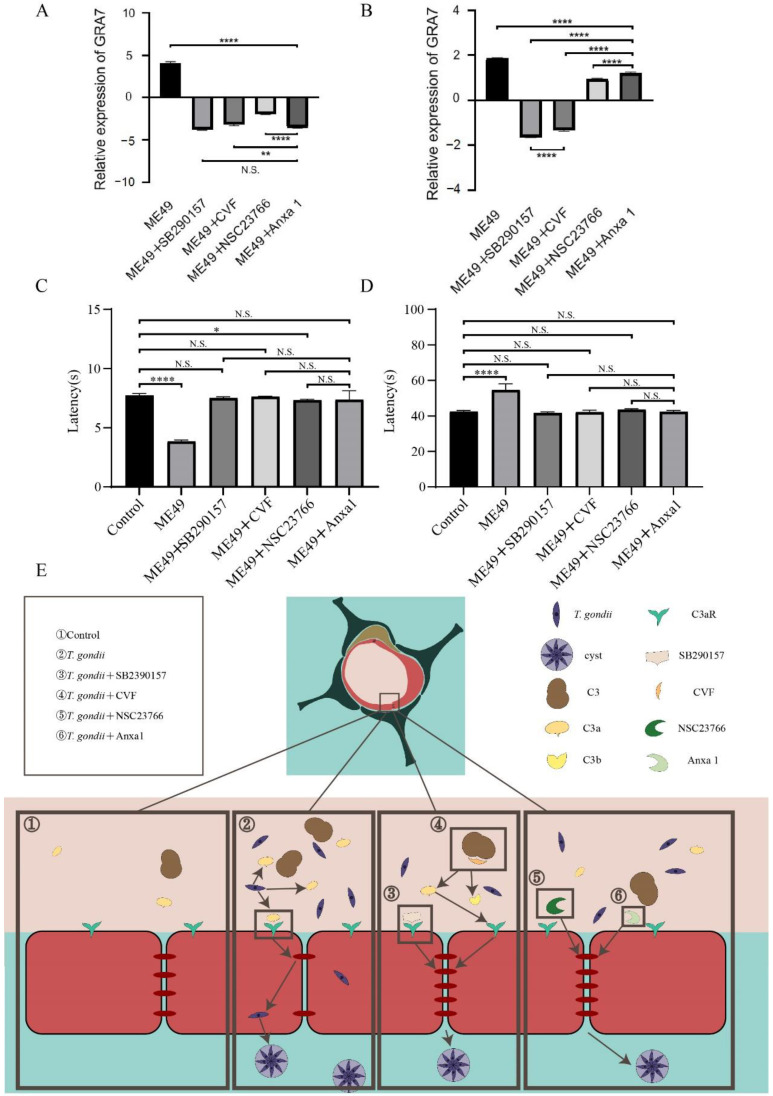
Four chemotherapeutic compounds inhibited *T. gondii* from entering the brains of mice and impairing their mental cognition. (**A**,**B**) The comparison of the expression of GRA7 in brain samples (**A**) and blood samples (**B**) of mice infected with ME49 and treated with four different chemotherapeutic compounds. The relative expression used β-actin to normalize. (**C**,**D**) The time comparisons obtained through rotating rod instrument experiment (**C**) and Morris Water Maze assay (**D**) by mice infected with ME49 and treated with different chemotherapeutic compounds. Statistical differences were analyzed by one-way ANOVA. N.S. *p* ≥ 0.05, * *p* < 0.05, ** *p* < 0.01, **** *p* < 0.0001. (**E**), Schematic model of four chemotherapeutic compounds inhibiting *T. gondii* from entering host brain tissue through the BBB.

**Table 1 molecules-27-05572-t001:** The primer sequences of C3a, GRA7, iNOS, ZO-1, Claudin-5, and β-Actb used in quantitative PCR.

Genes	Forward (5′→3′)	Reverse (5′→3′)
C3a	ATGACTCCCAGCACAAAGGG	CTCTCTTGCGGACCATCTCC
GRA7	CTAACCACCGGCCAGAATGT	TACGTCCTCGTGAGACCCAT
iNOS	AGCTATGTGACCACGTCCAC	CACCGGGGGATGAGGAATAG
ZO-1	TCTTGCAAAGTATCCCTTCTGT	GAAATCGTGCTGATGTGCCA
Claudin-5	GTTAAGGCACGGGTAGCACT	CAACGATGTTGGCGAACCAG
β-Actb	GGCTGTATTCCCCTCCATCG	CCAGTTGGTAACAATGCCATGT

## Data Availability

The data presented in this study are available in this article.

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
