# Peer review of "Four Chemotherapeutic Compounds That Limit Blood-Brain-Barrier Invasion by *Toxoplasma gondii"

_molecules, 2022, doi:10.3390/molecules27175572_

Round 1
Reviewer 1 Report
The present paper is an interesting contribution towards the dynamics of cyst formation during the chronic stage of T. gondii in the nervous system seraching for the potential key targets for new therapeutic drugs against toxoplasmic encephalitis.
The manuscript is conceptually and technically sound compilling a consistent work. It is quite well written and supported by nicelly designed comprehensive tables and figures making it appealing and ease to be red and followed.
However the writting of some sections of the results should be thoroughly revised, i.e. lines 275 and 298-99.
Also Toxoplasma gondii, should be typed in italics throughout the whole manuscript.
Furthermore the effect of four chemotherapeutic medications for cognitive behavior disorders in mice carrying brain T. gondii infections was also tested by rotation road instrument experiments. The results clearly indicate that the 4 medications could amend the cognitive impairments in mice due to the inhibition of T. gondii brain settlement. However, some of them (v.g. Anxa1) had no inhibitory effect on the parasite proliferation in blood. Therefore further insights on the mechanisms by which Anax1 is invading the brain barrier as well as the side effects are still pending.
In summary I advise the publication of this paper after some revision.
Minor points:
Line 47: The term "autoimmune" does not seem to be proper in this context.
Line 92: Delete the term "is" between the words "syntase" and "often"
Ln 115: Insert "in vitro" after "during".
Ln 135: The samples "were" placed in stead of "was".
Lns 146-148: The proteins "were" observed.....
Ln 154: E. coli must be in "italics"
Lns 161-162: The resin..........lysis buffer. "It was" finally eluted.....
Ln 171: Coverslips "were" stained........
Ln 338: "Plasmodium" must be replaced by "Toxoplasma"
Reviewer 2 Report
This manuscript investigates the effect of inhibitors of the C3 complement pathway on T. gondii infection of brain endothelial cells in vitro and in vivo, in brain tissues of infected mice.
A major issue is that authors widely used the term ‘Toxoplasmic encephalitis’ (TE) in the title, abstract, and throughout the text. However, the model of infection used in this study does not show that TE is induced by infection, so it is very speculative to claim that the molecules investigated here can inhibit TE development in absence of such a model or methodologies that support TE development (inflammation, immune cell recruitment).
Some other points need to be addressed:
- Results do not show any effect of inhibitors on mouse behaviour.
- The rationale for investigating GRA7 is poorly discussed
- Some results are not clearly presented in figures (e.g. Figure 5E)
- It is hard to follow results and figure panels that sometimes mix in vitro and in vivo results (as Fig 1). Figure legends are sometimes minimalist (e.g. Fig 5)
- I am not convinced by SEM micrographs, which look like TEM
- Anxa 1 is not a discovery (L. 98) as it is annexin A1
- English must be checked again
L 40 : protozoan
L 40 : most studies report that ~30% people are infected worldwide. The range is wider than 30-50% as seroprevalence is nearly less than 10% from certain areas to almost 100% in others.
L 42: caution. Several studies have shown that oocyst-related infections could be more symptomatic than those due to cyst ingestion. I suggest to delete this part of the sentence as it is not relevant in the scope of the present study.
L 44: I don’t know what ‘neurophilic by nature’ means. Does it mean that there are some molecular mechanisms that preferentially drive the parasite to the BBB? I am not sure. Toxoplasma parasites also migrate to muscles and other tissues (eyes), not only brain.
L 47: do you mean the host immune system? (not autoimmune, which is more related to autoimmune diseases)
L. 51: lead to TE is better than reactivate TE
Paragraph L 50-61: Agree that there is an increasing number of studies that report association between T. gondii infection (seroprevalence) and mental diseases or disorders. It is not necessarily linked to TE. So, it is not clear from what authors state here whether the rationale of this study is to find new drugs to treat TE, chronic infection (which is different from TE), or new drugs to clear tissue cysts from brain tissues, or new drugs to treat mental illness that could be associated to T. gondii infection. Please rewrite this paragraph.
L 65: delete ‘and so on’.
L. 65-66: do the authors mean C3 convertase instead of C3 invertase? Please clarify the terminology
L 74: mouse brain tissues
L. 98: Anxa 1 = Annexin A1? If so, this is not a discovery as stated line 98.
L. 99-100 and throughout the manuscript: the term ‘medications’ is not appropriate as some of the molecules used here to interfere C3 pathways (NSC23766, anxa1, CVF) are not used to treat patients. Inhibitors is a more appropriate term at least for NSC23766 and CVF. Whether Anxa1 is annexin A1 or a new protein is not clear.
L. 107: what do you mean by ‘suitable environment’?
L.114: what is normal saline? NaCl 0.9%?
L. 116-117: In bioassays, were the drugs injected before/concomitantly to/ after the parasites?
L. 122: specify the cell type of bEnd3 cells
L. 125-126: How many cysts per mouse?
L. 129: it is not clear what C3 Elisa detects? C3a production in brains?
L. 147-148: is it a part of sentence? Verb lacking. Perform.
L. 154: hyphenation trouble with Escherichia coli. Check your word processor.
L. 178: mouse brain tissues
L. 186-196: For those not familiar to such assays, it is not clear what paragraph corresponds to Morris water maze test.
L. 200 and L. 264: italicize T. gondii. Specify groups of mice.
L. 203 and Figure 1B legend: it looks like TEM micrographs rather than SEM. Are they brain sections in Fig 1B? Moreover, please add a scale bar to Fig1B panel.
Fig 1C: how did the gray values determine?
L. 217: specify that it is expression of TJ proteins
Paragraph 3.2: I guess that except for the control group, all groups were infected by Me49 parasites, and then some of them received C3a or the SB290157. Is it correct?
L. 235: adding low concentrations of C3a increased C3a expression. How do the authors explain that?
L. 248: TJ expression
Fig 3: please add scale bars
L. 284 and 286: again not ‘chemotherapeutic medications’; compounds; mouse brain
L. 288: what was the rationale for GRA7 detection?
L. 298-308 and Figure 5C-D: Results shown in Fig 5C-D do not support the text L. 298-308. Statistics from ME49 group compared to M49+each of the inhibitors should be presented.
Figure 5C and D: I don’t understand what is ‘Latency(s)’ as y-axis legend
Figure 5C and D: it is quite unsual to report p values < 0.0001 (****); usually cutoff is p<0.001 (***)
Figure 5E: there is no description of what Fig 5E represents. Moreover, TJ proteins are not formally reported, while this manuscript claims modifications of TJ protein expression. From this scheme, parasites should also be represented as moving between cells because of rupture of TJ and BBB integrity. How this model fits with other models published recently?
L. 322: TJ proteins
L. 330-334: ok, authors explain why C3a rises during infection, but do not explain why C3a decreases at high MOI.
L. 335: this sentence is very vague. It could be applied to a lot of pathogens!
L. 338: Studies on IFN-g and NO relationships on T. gondii should be mentioned before to visit Plasmodium studies
L. 338: Arginine can
L. 344: a peak of what?
L. 354: I understand now that Anxa1 is annexin A1. So please revise the claim that a new protein was discovered in this study (see L. 98), which is incorrect.
L. 376-379: it is not supported by the results, and so very speculative.
L. 389: T. gondii breeding?
L. 391 and 392: Anax1?
L. 394-397: I don’t understand this paragraph. It appears out of the discussion.
Reviewer 3 Report
All comments and suggestion are in the attached file

Round 2
Reviewer 3 Report
See the attached file
